# Proteogenomics of Colorectal Cancer Liver Metastases: Complementing Precision Oncology with Phenotypic Data

**DOI:** 10.3390/cancers11121907

**Published:** 2019-12-01

**Authors:** Bernhard Blank-Landeshammer, Vincent R. Richard, Georgia Mitsa, Maud Marques, André LeBlanc, Laxmikanth Kollipara, Ingo Feldmann, Mathilde Couetoux du Tertre, Karen Gambaro, Suzan McNamara, Alan Spatz, René P. Zahedi, Albert Sickmann, Gerald Batist, Christoph H. Borchers

**Affiliations:** 1Leibniz-Institut für Analytische Wissenschaften-ISAS-e.V., Otto-Hahn-Str. 6b, 44227 Dortmund, Germany; bernhard.blank@isas.de (B.B.-L.); Laxmikanth.kollipara@isas.de (L.K.); ingo.feldmann@isas.de (I.F.); Albert.sickmann@isas.de (A.S.); 2Segal Cancer Proteomics Centre, Lady Davis Institute, Jewish General Hospital, McGill University, Montreal, QC H3T1E2, Canada; vincent.richard@ladydavis.ca (V.R.R.); georgia.mitsa@mail.mcgill.ca (G.M.); andre.marc.leblanc@gmail.com (A.L.); 3Exactis Innovation, 5450 Cote-des-Neiges, Suite 522, Montreal, QC H3T1Y6, Canada; MMarques@exactis.ca (M.M.); MCouetoux@exactis.ca (M.C.d.T.); KGambaro@exactis.ca (K.G.); SMcNamara@exactis.ca (S.M.); gerald.batist@mcgill.ca (G.B.); 4Segal Cancer Centre, Lady Davis Institute, Jewish General Hospital, and McGill University, Montreal, QC H4A3T2, Canada; alan.spatz@mcgill.ca; 5Medizinische Fakultät, Medizinisches Proteom-Center (MPC), Ruhr-Universität Bochum, 44801 Bochum, Germany; 6Department of Chemistry, College of Physical Sciences, University of Aberdeen, Aberdeen AB243FX, UK; 7Gerald Bronfman Department of Oncology, Jewish General Hospital, McGill University, Montreal, QC H4A3T2, Canada

**Keywords:** proteogenomics, KRAS, targeted mass spectrometry, PRM, absolute quantitation, mutation rates

## Abstract

Hotspot testing for activating *KRAS* mutations is used in precision oncology to select colorectal cancer (CRC) patients who are eligible for anti-EGFR treatment. However, even for *KRAS^wildtype^* tumors anti-EGFR response rates are <30%, while mutated-*KRAS* does not entirely rule out response, indicating the need for improved patient stratification. We performed proteogenomic phenotyping of *KRAS^wildtype^* and *KRAS^G12V^* CRC liver metastases (mCRC). Among >9000 proteins we detected considerable expression changes including numerous proteins involved in progression and resistance in CRC. We identified peptides representing a number of predicted somatic mutations, including KRAS^G12V^. For eight of these, we developed a multiplexed parallel reaction monitoring (PRM) mass spectrometry assay to precisely quantify the mutated and canonical protein variants. This allowed phenotyping of eight mCRC tumors and six paired healthy tissues, by determining mutation rates on the protein level. Total KRAS expression varied between tumors (0.47–1.01 fmol/µg total protein) and healthy tissues (0.13–0.64 fmol/µg). In *KRAS^G12V^*-mCRC, G12V-mutation levels were 42–100%, while one patient had only 10% KRAS^G12V^ but 90% KRAS^wildtype^. This might represent a missed therapeutic opportunity: based on hotspot sequencing, the patient was excluded from anti-EGFR treatment and instead received chemotherapy, while PRM-based tumor-phenotyping indicates the patient might have benefitted from anti-EGFR therapy.

## 1. Introduction

In 2018, 1.8 million people worldwide were diagnosed with colorectal cancer (CRC) [1], and 860,000 people died from CRC, making it the third most common and second most deadly cancer worldwide [2]. An increased risk for CRC can be linked to certain lifestyle factors such as physical inactivity, obesity, smoking, alcohol, and specific dietary preferences [3]. Approximately 39% of CRC patients present with localized disease (5-year survival of 90%). While CRC can often be cured when diagnosed early, 50–60% of patients diagnosed with CRC will develop metastases during the course of their disease [4]. The 5-year survival rate for patients with distant metastatic disease, i.e., where the cancer has spread to the liver, lungs, or distant lymph nodes (includes Stage IV cancers), is only 14% [5]. Treatments for metastatic CRC (mCRC) are still largely based on toxic chemotherapy regimens, but important insights into tumor biology enabled the development of targeted cancer therapies. Because 60–80% of CRC tumors show epidermal growth factor receptor (EGFR) overexpression, which is associated with a poor prognosis in patients [6,7], anti-EGFR antibodies—such as Cetuximab [8] or Panitumumab [9] are important targeted therapies in CRC that contributed to the improved median survival, which increased from 5 months to 2 years between 1993 and 2009 [10]. Interestingly, new therapeutic strategies incorporating EGFR inhibitors are emerging as promising treatments in patients whose cfDNA harbors *EGFR* amplifications [11]. An important determinant of whether patients are eligible for anti-EGFR therapies is their *KRAS* mutational status, which has become a validated predictor of non-response to anti-EGFR antibodies [8]. The biological rationale is that the most frequently observed mutations activate KRAS transcription, so that the downstream MEK/ERK signalling pathway is constitutively active, making these cells insensitive to the antibodies blocking the upstream ligand binding site. It has been demonstrated that *KRAS^WT^* patients benefit from cetuximab, whereas *KRAS^mut^* patients very seldom do [12,13]. Other putative biomarkers, such as EGFR ligands, have generated conflicting and inconclusive results_,_ so *KRAS^mut^* remains the only biomarker in clinical use [14,15]. Consequently, it has become clinical practice in precision oncology to check the *KRAS* mutational status to avoid treating patients with predictably ineffective drugs, and this has also led to significant reduction in treatment cost. Nevertheless, of those *KRAS^WT^* patients who receive anti-EGFR therapies, <30% actually respond [13], indicating an urgent need for better predictive biomarkers.

Modest response rates in precision oncology can, for instance, arise from therapeutic resistance due to the activation of alternative signalling pathways. This has been demonstrated for bevacizumab, where vascular endothelial growth factor (VEGF) inhibition can trigger signalling through Insulin-like growth factor 1 receptor (IFG1R), platelet-derived growth factor receptor (PDGFR), Fibroblast growth factor receptor (FGFR), or hepatocyte growth factor receptor (MET) [16]. Predicting the actual pathway activity on the protein level would be an important step forward to better choose therapeutic options and overcome resistance. However, this cannot be readily accomplished using genomics data. This inconsistency between genomics data and the actual phenotype can be attributed to a variety of causes:

(i) Genomics/transcriptomics data lacks information on translational (protein synthesis and degradation) and posttranslational (e.g., protein activity) control of pathway activity [17]. (ii) It has been demonstrated that mRNA levels do not reliably predict protein abundances [18]. (iii) Many genomic abnormalities may not be transcribed and translated into proteins [19]. (iv) Translation of unexpected areas of the genome, non-canonical reading frames, and post-transcriptional events may lead to unexpected protein products [18,20]. These are critical points, because proteins are the targets for the vast majority of therapeutic agents.

One strategy for improving current precision oncology approaches for better targeted-therapy prediction is to improve the phenotyping of individual tumors by complementing current genome-based approaches with mass spectrometry data on actual protein expression and post-translational modifications (PTMs)-i.e., proteogenomics. As demonstrated by the clinical proteomic tumor analysis consortium (CPTAC), only the integration and clustering of DNA, RNA, protein, and protein phosphorylation profiles allowed distinguishing subtypes in 77 breast cancer tumors [21]. In another proteogeonomics study, Huang et al. applied quantitative (phospho)proteomics to study 24 breast cancer-derived xenografts (PDX) models [22] and not only confirmed the predicted genomic targets, but also found protein expression and phosphorylation changes that could not be explained based on genomic data alone. Recently, CPTAC reported a CRC proteogenomics study where they analyzed primary tumors and matched healthy tissues from 110 CRC samples [23]. In a major effort, this study correlated increased retinoblastoma protein (RB1) phosphorylation levels with increased proliferation and decreased apoptosis in CRC and suggested that glycolysis is a potential target for overcoming the resistance of micro-satellite instability-high tumors to immune checkpoint inhibitors.

Here, we describe a proteogenomic analysis of CRC liver metastases (metastatic CRC, mCRC; Figure 1a–e), an ideal setting for the analysis of therapeutic resistance which occurs in a short timeframe, and the clinical context for almost all clinical testing of novel therapeutics. Biopsies from liver metastases were collected from two mCRC patients after relapse on first-line treatment, and both whole exosome sequencing (WES) and RNAseq data was made available for these specimen by Exactis Innovation (Clinicaltrials.gov NCT00984048). We demonstrate how targeted mass spectrometry can be used to determine mutation rates on the protein level and how this may help to address the discordance between KRAS mutational status and response rates to anti-EGFR treatment in precision oncology.

## 2. Results

### 2.1. Depth of the Proteome

Deep-proteomic profiling resulted in the identification (based on at least one unique peptide) of 8603 and 8330 unique proteins, at 1% protein false discovery rate, from *KRAS^WT^* and *KRAS^G12V^* tumors, respectively (Appendix A). We calculated the normalized spectral abundance factors (NSAF), developed by Washburn and coworkers [24], to estimate the relative abundance of proteins within the tumors and to also allow a quantitative comparison (Appendix A). NSAF has been previously shown to produce robust quantitative spectral-counting data, even allowing the reliable estimation of copy numbers per protein if calibrated using absolute quantitative reference values within a sample [25,26]. For both samples, we covered five orders of magnitude of dynamic range of the proteome (Figure 2a) with complete coverage of important cancer pathways, including 458 transcription factors [27], 318 kinases/kinase subunits [28], 157 phosphatases/phosphatase subunits [29], and 345 proteases [30]. We detected 9 and 11 components of the mitochondrial import complexes TOM and TIM, respectively, including small proteins such as translocase of inner mitochondrial membrane 8 (TIMM8), TIMM9, TIMM10, TOMM5, translocase of outer mitochondrial membrane 6 (TOMM6), and TOMM7, demonstrating a good coverage of membrane proteins in general. Our data has a nearly complete coverage of important cancer signaling pathways, such as shown for the PI3K/AKT/mTOR pathway, which is the primary target in targeted CRC therapy (Figure 2b).

In addition, we identified a total of 4144 unique high-confidence (<1% FDR; phosphoRS > 0.9) phosphorylation sites from 4888 phosphopeptides, 763 of which were at least 4-fold differentially regulated between the two tumors, based on spectral counting. The differential phosphoproteins were significantly enriched in REACTOME [31] pathways-for instance, those related to SUMOylation (4.45 × 10^−11^), signaling by Rho GTPases (1.10 × 10^−6^), and mRNA splicing (2.00 × 10^−3^). Among the proteins showing increased levels of phosphorylation in the *KRAS^G12V^* sample were MEK and ERK, reflecting the expected activation of the RAS/MEK/ERK pathway upon the presence of the activating G12V-mutation in *KRAS* in this tumor (Figure 2b).

### 2.2. Quantitative Comparison of KRAS^G12V^ and KRAS^WT^ Tumors

By nature, tumor samples are heterogenous. Even under controlled conditions where clear standard operating procedures (SOPs) are strictly followed for sampling and sample preparation (as was done for the tumors analyzed in-depth in this study), a considerable amount of preanalytical variability cannot be excluded. The reasons for this include differing levels of tumor-infiltrating immune cells, necrosis, or simply the tumor content. We therefore decided to compare the proteomes of the *KRAS^WT^* and *KRAS^G12V^* tumors with deep-proteome profiling data from 19 different human (primary) cells and tissues (Figure 2c) which represented potential sources of contamination such as plasma, platelets, B-Cells, T-cells, connective and muscle tissue, and also HepaRG cells (liver stem cells), primary hepatocytes, as well as lymphocytes and megarkaryocytes as hematopoietic progenitor cells. We thus compared the expression patterns for a total of 9408 unique proteins identified in our tumor samples with these 19 different reference proteome samples (Appendix A). Importantly, Eucledian clustering showed that the CRC tumors represent a distinct cluster with the nearest neighboring cluster comprising liver and kidney stem cells, while blood contaminants and immune cells are further apart.

Next, to further identify whether different levels of contamination with blood, liver or immune cells might lead to artifactual proteomic differences between the tumor samples, we compared proteins showing a minimum fold-change of 4.0 between the *KRAS^WT^* and *KRAS^G12V^* tumors with the top 100 proteins (based on NSAF values) of each of our 19 reference proteomes (Appendix A). Our rationale was that if a substantial contamination of one of the tumor samples by, for example blood, immune cells, or healthy liver tissue, had led to the false-positive identification of differential tumor proteins, it would be accompanied by a considerable enrichment of the top 100 proteins from the respective contaminant. In fact, we found no major source of preanalytical variability among the 800 proteins that were at least 4.0-fold regulated between *KRAS^WT^* and *KRAS^G12V^*. Gene ontology analysis using PANTHER [35] showed an enrichment of cell migration, (*p*-value 1.85 × 10^−5^), regulation of the MAPK cascade (7.99 × 10^−5^), and transcription by RNA polymerase II (1.60 × 10^−4^) in the *KRAS^G12V^* tumor, while no significantly enriched pathways were detected in the *KRAS^WT^* sample. In general, the two proteomes of *KRAS^WT^* and *KRAS^G12V^* show a similar expression of pathways and biological functions (Figure 3).

Of the 800 proteins that were differentially regulated between the tumors, 31 had somatic mutations detected by WES and/or RNAseq, and 596 transcripts showed the same trend of regulation in the RNAseq data. Interestingly, the average fold-change for those 31 mutated proteins was >12, indicating a major impact of those mutations on protein expression. Transcriptome and proteome data only poorly correlate for both tumors, WT and G12V (Figure 3d,e). The G12V/WT ratios derived from either the proteome or the transcriptome also correlate poorly (Figure 3f). Among the 250 proteins with the strongest discordance between proteome- and transcriptome-derived G12V/WT ratios were 8 transcription factors, 4 kinases, 2 phosphatases, 18 proteases, 10 histones, several members of the WNT-signaling pathway (WNT2, FRZB, SFRP4), the ADP-ribosylation factors ARF1 and ARF3, MUC2, CYP1B1, the SRC signaling inhibitor SRCIN1, eIF5A2, and LAMA2.

Mucin-2 (MUC2), for which the T1636S single nucleotide variant (SNV) was detected by WES in the G12V sample, showed the strongest protein upregulation in G12V (MUC2↑G12V; 1 unique peptide and 2 peptide-spectrum matched (PSMs) in the WT vs. 97 unique peptides and 2361 PSM in the G12V), while it was upregulated to a much lesser degree on the transcriptome level (7.4-fold upregulated in G12V RNAseq vs. WT RNAseq data). It has been reported that patients with low expression (here, WT) of MUC2 had significantly lower cell differentiation and more lymph node metastases than those with higher MUC2 levels (G12V) [37]. In contrast, high expression of Mucin-5 (MUC5) was associated with lymph node metastasis and poor cellular differentiation and poor prognosis [37]. Indeed, MUC5B was also clearly upregulated in the G12V tumor proteome (MUC5B↑G12V; 4 peptides and 8 PSM in WT vs. 17 peptides and 56 PSM in G12V; 5.6-fold higher in the G12V transcriptome), and the reduced expression of MUC5B in the WT might derive from a N1032I SNV that was detected in its WES data. Interestingly, MUC12 was also found to be clearly higher in the G12V tumor both on the proteome and the transcriptome levels (MUC12↑G12V; absent in WT, 26 peptides and 76 PSM in G12V; 13.0-fold higher in G12V RNAseq) and was found to be mutated in both tumors, with the SNV V874I in the G12V (both WES and RNAseq) and the SNV V3487I in the WT (only WES). MUC12 expression has been identified as a marker of prognosis in Stage II and III CRC, resulting in a worse prognosis at low MUC12 expression in qPCR [38].

Another protein that showed strong regulation between both tumors was Laminin subunit alpha-2 (LAMA2↑G12V; 17 peptides and 39 PSM in WT vs. 114 peptides and 464 PSM in G12V), for which an A1805S SNV was detected by WES in the G12V. Interestingly, LAMA2 was not differentially regulated on the mRNA level (ratio G12V/WT of 0.84), and is a very strong example for discordance between transcriptome and proteome. LAMA2 is a suggested tumor suppressor [39] and is frequently mutated in other cancers, such as lung cancers [40]. Indeed, LAMA1 was not detected in the WT tumor while being moderately expressed in the G12V tumor (LAMA1↑G12V), and the transcriptome data showed a 4.3-fold higher expression level in the G12V tumor.

The ADP-ribosylation factor 1 (ARF1) has been reported as significantly elevated in various cancers [41], and its expression in prostate cancer correlated with activation of ERK1 and ERK2, leading to cell proliferation [42]. We found ARF1 to be significantly higher in the WT tumor proteome (ARF1↑WT; no peptides/PSM in G12V vs. 10 peptides and 137 PSM in WT), which might represent an EGFR-independent activation of the RAS/MEK/ERK pathway and a potential escape mechanism for anti-EGFR therapy. The strong regulation on the protein level is in stark contrast to the transcriptome data, which indeed show a slightly lower ARF1 mRNA level in the WT tumor (1.3-fold higher in the G12V transcriptome data). Screening for ARF1 in association with mutations in KRAS could be an enhanced predictive signature for anti-EGFR therapy, but clearly ARF1 is preferentially measured on the protein level.

Receptor tyrosine-protein kinase erbB-2 (ERBB2/HER2) showed a massively higher protein expression in the WT than in the G12V tumor (ERBB2↑G12V; 46 peptides and 907 PSM vs. 4 peptides and 12 PSM), in accordance with a detected copy number gain (amplified region chr17:37,690,344–40,762,015) and 69-fold higher normalized counts in the RNAseq data of the WT tumor. ERBB2/HER2 expression in the WT tumor was also significantly higher than in any of our 19 reference proteomes. The opportunity to target ERBB2/HER2 in colorectal cancer has recently emerged [43] as it is amplified and/or mutated in 5% of CRC tumors, most often in KRAS WT tumors [43], which agrees with our data.

Mesothelin (MSLN) has recently been described as a prognostic marker for Stage II/III CRC where its expression was associated with a lower survival rate. We could not detect MSLN in the G12V tumor proteome, while it was highly expressed in the WT tumor (MSLN↑WT; 22 peptides and 118 PSM) which is in good agreement with the transcriptomic data.

Another protein that has been associated with a poor outcome in Stage II and III CRC is the serine protease HTRA3 which was not detectable in the WT tumor proteome, but clearly elevated in the G12V (HTRA3↑WT; 19 peptides and 68 PSM). The transcriptome data showed the same trend (5.3-fold up-higher mRNA level in the G12V tumor).

Gremlin-1 has been associated with improved survival in locally advanced Stage II and III CRC [44], and could not be detected in the WT tumor proteome but was highly expressed in the G12V tumor (GREM1↑G12V) and showed an 11.1-fold higher mRNA level in the G12V tumor, in accordance with the proteome data.

Furthermore, eukaryotic translation initiation factor 5A-2 (eIF5A2) was not detected in the WT tumor but is highly expressed in the G12V tumor (eIF5A2↑G12V). eIF5A2 has been shown to promote chemoresistance to doxorubicin in CRC [45]. Its overexpression in the non-anti-EGFR eligible G12V tumor might have been an important implication for chemotherapy. The eIF5A2 mRNA level was also higher in the G12V transcriptome (2.0-fold), but to a much lower extent.

Taken together, our deep-proteome profiling of the *KRAS^WT^* and *KRAS^G12V^* tumors revealed a number of important expression changes (Figure 3), some of which might have a direct application to therapeutic options. Clearly, these phenotypic insights on actual protein expression levels are an extremely valuable complement to the precision oncology data on individual tumors.

### 2.3. Identification and Quantification of SNV Predicted from the WES and RNAseq Data

A total of 51- and 77-point mutations in the *KRAS^WT^* and *KRAS^G12V^* samples were predicted by both WES and RNASeq, while 316 and 384 were predicted only by WES, with no evidence from RNASeq (Appendix A). We used this information to conduct mutation-directed database searches in order to identify peptides in our deep proteome profiling dataset that represent evidence for the presence of those mutations on the protein level. Indeed, we were able to identify a total of three and seven mutations in the *KRAS^WT^* and *KRAS^G12V^* samples, respectively, with high confidence at 1% group-specific FDR, as had been proposed by Nesvizhskii [46]. For all of these mutations, we could also identify the corresponding canonical sequences in the same tumor sample.

We hypothesized that targeted mass spectrometry could be utilized to quantify the actual mutation rates on the protein level, and thus enable an improved phenotyping of individual tumor samples by precise quantification of both mutated and canonical variants of target proteins. We therefore synthesized stable-isotope labeled standards (SIS) for both the mutated and the canonical variants of eight mutated proteins (Table 1), and developed corresponding parallel reaction monitoring (PRM) assays. Among our candidates was KRAS^G12V^. Notably, the canonical sequence LVVVGAGGVGK is shared between KRAS/HRAS/NRAS, so quantification of this peptide would not allow the determination of the G12V mutation rate. We therefore added unique peptides that represent HRAS, NRAS, and KRAS. We evaluated our PRM assays using the *KRAS^G12^* and *KRAS^WT^* samples, and were able to achieve absolute quantification for KRAS from as little as 3 µg of total tissue protein digest on-column, using nano-LC-PRM.

To further evaluate the utility of our assays, we obtained additional mCRC liver-metastasis samples (T1–T6) and matched healthy liver tissue (H1–H6) from another six patients with *KRAS^G12V^*-positive tumors, as defined by hotspot mutation testing for the presence of selected mutations. Notably, these samples were not part of a well-designed study, but were ‘real-life’ biopsies from our internal biobank and therefore represent the ideal setting to evaluate whether targeted MS can be used to improve the phenotyping of samples that were not collected using specific SOPs for proteome analysis. All these patients were ineligible for targeted anti-EGFR treatment and received different (combinations) of treatment, including anti-VEGF-A treatment with Bevacizumab as well as chemotherapy for several cycles. In line with the poor prognosis of patients with Stage III and Stage IV CRC, four out of the six patients from the biobank have passed away. Starting with 50 µg of total protein for sample preparation and loading 3 µg of total tissue protein digest on-column, we quantified the eight proteins shown in Table 1 and their mutation rates.

We were able to quantify SRPX2 in all eight tumor samples (90–860 amol/3 µg; median 470 ± 260 amol) but in none of the six healthy control tissues, indicating a clear upregulation of the protein in CRC, in agreement with previous reports that SRPX2 plays a role in the progression of CRC [47]. The mutation rate of SRPX2^E234K^ was 66% in the *KRAS^WT^* sample, while the mutated protein could not be detected in any of the G12V tumors.

We were also only able to quantify RPS6KA5 in seven tumor samples (G12V, WT, T1-T5), where its concentration ranged from 110–680 amol/3 µg (median 370 ± 340 amol), but not in any of the healthy tissues. Interestingly, survival data from CRC patients in the Human Protein Atlas [48] indicate a better prognosis with high expression of RPS6KA5. The only sample where the RPS6KA5^D554N^ SNV could be detected was the *KRAS^WT^* tumor with a mutation rate of 58%.

PTBP1 could be quantified in all samples, ranging from 2.7–21.2 fmol/3 µg in the tumor samples (median 12.7 ± 7.2 fmol/3 µg) and 3.7–9.1 fmol/3 µg in the controls (median 7.4 ± 2.4 fmol/3 µg), showing no clear trend of regulation between cancer and control tissue. The mutated variant PTBP1^K508E^ could only be detected in the original *KRAS^G12V^* and in tumor T6, amounting to 38% and 26% mutation rates, respectively.

ARL2 could be quantified in all samples, with mutation rates between 35–100% in the tumors (median 620 ± 180 amol/3 µg) and 52.7–100% in the control tissues (median 170 ± 50 amol/3 µg). ARL2^V141A^ is a natural variant with frequencies between 0.5 and 0.7 in various cohorts. The presence of the mutation in all eight patients of this small cohort might be random, but it might be important to follow this up in a larger study.

PPP1R14C, an inhibitor of the Serine/threonine-protein phosphatase PP1-alpha catalytic subunit (PPP1CA), could only be quantified in five tumors (*KRAS^WT^*, *KRAS^G12V^*, T1, T3, T6), but in none of the healthy controls, again indicating a potential upregulation of the protein in a mCRC setting (median 70 ± 20 amol/3 µg; 530 amol in G12V). Interestingly, in all five tumors, only the mutated variant PPP1R14C^T10A^ could be detected, notably a natural variant with frequencies up to 0.8 in East Asian cohorts.

HAUS7 showed a 100% mutation rate in the *KRAS^WT^* and the T1 tumor sample, but no consistent trend between the tumors (median 180 ± 10 amol/3 µg) and the paired healthy controls (median 80 ± 20 amol/3 µg) was observed.

TBC1D2B^A8G^ could be detected in two G12V tumors (*KRAS^G12V^* 49%, T4 100%). Notably, this mutation was also detected in healthy control H4 at a mutation rate of 100%.

Figure 4 summarizes the results for KRAS for the eight tumors and six controls. Total KRAS expression in the tumor samples varies between 2.0 and 5.7 fmol/3 µg (median 3.0 ± 1.4 fmol/3 µg), and between 0.4 and 1.9 fmol/3 µg (median 1.6 ± 0.6 fmol/3 µg) in the healthy control tissues. KRAS was upregulated in all tumors (T1–T6) compared to their healthy controls (H1–H6), by between 1.3-fold in T4/H4 and 14.8-fold in T1/H1. The strong upregulation of KRAS in T1 (at a mutation rate 86%, in stark contrast to the WES predicted frequency of 32%) might indicate a massive activation of the MEK/ERK signaling pathway in this patient’s tumor. While the G12V SNV could not detected in any of the healthy controls H1–H6 nor in the *KRAS^WT^* tumor, the mutation rate for the G12V positive tumors varied considerably (*KRAS^G12V^* 50%, T1 86%, T2 100%, T3 52%, T4 38%, T5 10%, and T6 42%). The surprisingly low mutation rate of only 10% in T5 may indicate that this patient might be one of the false-negative patients who might benefit from targeted anti-EGFR treatment despite the detection of the *KRAS^G12V^* mutation during hotspot sequencing.

### 2.4. Robustness of NSAF-Based Relative Quantification

Finally, to validate the NSAF-based proteome-wide quantification, we compared the ratios between the *KRAS^G12V^* and *KRAS^WT^* tumors obtained from PRM-based protein quantification using SIS peptides with those obtained from our NSAF values. Notably, TBC1D2B was not detected by PRM analysis of the WT sample (NSAF-based G12V/WT ratio: 1.4). Plotting the NSAF against the PRM ratios yields a comparably low R^2^ of 0.87, mainly due to the outlier SRPX2 (PRM: 3.2; NSAF: 0.9). Without this outlier, plotting the NSAF against PRM-derived ratios leads to an R^2^ of 0.99 (Appendix A).

## 3. Discussion

Precision oncology has a great potential to help in the global battle against cancer, yet only a fraction of patients are eligible for genome-directed targeted therapies. It is estimated that, at best, only 8.3% of US patients would be eligible and that, at best, only 4.9% would respond to the treatment [50]. One of the reasons still is the lack of treatment options. Another reason, however, is the discordance between the genome and the proteome, the latter being a better representation of the actual tumor phenotype. Indeed, a major challenge in using genome-only based molecular profiling is to identify which, amongst sometimes multiple variants present, represents a driver of tumor growth and therefore should be specifically targeted. Given the energy a cell has to invest in producing a functional protein, we propose that over-expressed proteins represent molecules that give the tumor cells a growth advantage.

Proteogenomics can contribute to precision oncology in two major ways: (i) The systematic profiling of patient cohorts can help to identify novel treatment options, as recently demonstrated for primary CRC tumors [23]. (ii) The improved phenotyping of patient tumors can help to better understand not only the causes of unexpected resistance, but also unexpected responses to targeted therapies that cannot be explained based on existing genomic data.

In this study, we conducted a genomics-driven proteomic profiling of a *KRAS^G12V^* and a *KRAS^WT^* mCRC liver metastases. These tumors are associated with a poor prognosis and often are associated with resistance upon first-line treatment and are, therefore, important samples for studying the reasons for resistance and for identifying novel therapeutic windows. We identified almost 9000 unique proteins whose concentrations varied over 5 orders of magnitude, and more than 4000 high confidence phosphorylation sites. A good correlation (*R*^2^ = 0.99) between PRM-based quantification using SIS peptides and spectral counting data using the normalized spectral abundance factor (NSAF) for 7 ‘absolutely quantified’ proteins confirms that NSAF-based comparison of the deep proteome data is an adequate strategy for comparing *KRAS^WT^* and *KRAS^G12V^* tumors. NSAF-based proteomic clustering using a total of 19 reference proteomes representing different cancer cells, blood components, and primary tissues, confirmed the absence of major contaminants. Although this indicates a minimum impact of sampling-derived contamination on the proteomes, we focused on comparing 800 proteins having at least 4-fold differential regulation between the two tumors, to minimize the impact of the limitations in differentially quantifying low-abundance proteins using spectral counting. Despite a generally similar proteomic profile, the *KRAS^G12V^* tumor showed an enrichment of proteins involved in cell migration, MAPK cascade regulation, and RNA polymerase II transcription, and a differential pattern of phosphoproteins involved particularly in SUMOylation, Rho GTPase signalling, and mRNA splicing. For 32 proteins, we detected a major change in expression between *KRAS^G12V^* and *KRAS^WT^* that might be connected to somatic mutations identified in the corresponding WES/RNAseq data, while other proteins showed a very stable expression despite the presence of mutations in one of the samples. We found a number of differentially-expressed proteins that have been associated in the literature with (m)CRC and patient outcome, among both the mutated proteins and proteins without detected mutation in any of the mCRC liver tumors. Some of these changes were in line with RNAseq data, such as ERBB2/HER2 which was massively upregulated in both, the *KRAS^WT^* tumor proteome (also compared to all of our reference proteomes) and transcriptome as a consequence of a gene amplification (region chr17:37, 690, 344-40,762,015). Other proteomic changes where in stark contrast to the transcriptome data, such as LAMA2 which was more than 11-fold upregulated in the G12V proteome, but slightly downregulated in G12V transcriptome (G12V/WT 0.8). In accordance with other studies, we found a poor correlation between the proteome and transcriptome data.

## 4. Materials and Methods

### 4.1. Experimental Design and Statistical Rationale

The discovery-based deep proteomic profiling and phosphoproteomics single-shot analysis were conducted on two tumors, respectively *KRAS^WT^* and *KRAS^G12V^*, without technical replicates. Only unique peptides were considered, and only proteins identified with at least one unique peptide were reported and used to calculate the normalized spectral abundance factors (NSAF). Only phosphopeptides having a phosphoRS probability for site localization of at least 0.9 were considered.

The targeted parallel reaction monitoring (PRM) data was derived from two original *KRAS^WT^* and *KRAS^G12V^* mCRC liver metastases, as well as another 6 *KRAS^G12V^* positive mCRC liver metastases and their matched healthy tissues from the same patients. All measurements were done in technical triplicates. PRM data are represented as means of these three technical replicates.

### 4.2. Materials

The complete mini protease inhibitor, PhosSTOP phosphatase inhibitor, benzonase, ammonium bicarbonate (NH_4_HCO_3_), iodoacetamide (IAA), and guanidine hydrochloride (GuHCl) were purchased from Sigma Aldrich. Calcium chloride (CaCl_2_) was obtained from Merck, dithiotreitol (DTT) was purchased from Roche Diagnostics, and trypsin (sequencing grade) was obtained from Promega. Protein LoBind tubes were bought from Eppendorf. The Pierce BCA Protein Assay Kit was purchased from Thermo Fisher Scientific. Ultra-pure solvents (HPLC-grade)-i.e., formic acid (FA), trifluoroacetic acid (TFA), and acetonitrile (ACN)-were obtained from BioSolve. 

Gentra PureGene Blood kit was purchased from Qiagen for the buffy coat DNA extraction. The NanoDrop spectrophotometer and Qubit v2.0 Fluorometer from Thermo Fisher Scientific, Mississauga, ON, Canada, and the Agilent bioanalyzer 2100 from Agilent Technologies, Santa Clara, CA, USA were used to assess concentration, purity and degradation of nucleic acid extracts.

### 4.3. Sampling

Liver metastasis specimens were collected from two mCRC patients undergoing standard first-line therapy (NCT00984048), at baseline and at the time of disease progression as defined by RECIST 1.0. One patient received FOLFOX (fluro-pyrimidine 5-FU, folinic acid and oxaliplatin) and one patient received FOLFOX plus bevacizumab. Established SOPs were followed by site and laboratory personnel during biospecimen collection and processing [51,52,53,54]. Fresh-frozen samples were optimal cutting temperature compound (OCT) embedded to provide support during cryosectioning and to minimize lyophilization during long-term storage. Prior to OMICs analysis samples were macrodissected to isolate the tumor tissue. Cryosections (5 μm thick) were cut and stained with hematoxylin and eosin (H&E). The percentage of normal cells, tumor cells, and necrotic areas was determined and the threshold for suitable tumor cell area for extraction was set at >60% of the specimen including <20% of necrotic cells as suggested by the Cancer Genome Atlas (TCGA). When needed, biopsies were macro dissected to reach these thresholds. One representative sample per collection was selected for nucleic acid extraction. Samples with a ratio 260/280 > 1.8 and sufficient DNA quantity were used for WES profiling and samples with RIN > 3 were selected for total RNA sequencing. The internal code at the Jewish General Hospital for the specific proteogeonomics analysis is 2020-1752. All patients provided written informed consent for a biopsy of a liver metastatic lesion before treatment and another biopsy at disease progression. mCRC liver tumors and matched healthy liver tissue were obtained from the Jewish General Hospital Central Biobank. This biobank is affiliated to the Réseau de recherche sur le cancer (RRCancer) of the FRQS and to the Canadian Tumor Repository Network (CTRNet).

### 4.4. WES and RNAseq Data Generation

WES and RNAseq were performed on the samples at the McGill University and Genome Quebec Innovation Center, according to current standards in precision oncology [55]. Sequencing was done using an Illumina HiSeq 2500 using the ‘rapid-run’ mode with 100 bp paired-end reads. Reads were trimmed using Trimmomatic (v0.35) [56], removing the adaptor, the first four bases from the start of each read, and low-quality bases at the end of each read, using a 4-bp sliding window to trim where the average window quality fell below 30. Trimmed reads < 30 bp were discarded. The clean reads were then aligned to the reference genome hg19 (GRCh37) using BWA-MEM v0.7.13 [57]. Duplicated reads were marked and filtered out so that only unique DNA fragments were used in the subsequent analysis, which was done using Picard v2.1.0. Using the Genome Analysis Toolkit, potential Indels were identified with the RealignerTargetCreator, and reads were realigned in these targeted regions using IndelRealigner [58,59]. Somatic point mutation calling was done using MuTect2 using paired normal and tumor samples [60]. SNVs were annotated using ANNOVAR [61]. Total RNA was sequenced on a HiSeq 2500 (Illumina) with PE75 and with PE125 kits at Genome Quebec, Canada. Data was processed as above, and clean reads were aligned to the reference genome hg19 (GRCh37) using STAR [62]. Gene counts were obtained using featureCount [63] with custom gtf files. Counts were normalized using the library size.

### 4.5. Sample Preparation and Discovery Proteomics

#### 4.5.1. Tissue Homogenization and Protein Extraction

Snap-frozen tissue samples were ground in a glass Dounce homogenizer using 600 µL of lysis buffer (4% sodium dodecyl sulfate, 150 mM NaCl, 50 mM Tris, pH 7.8; supplemented with complete mini and PhosSTOP according the manufacturer’s recommendations). Homogenization was supported using an ultrasonic bath (3 × 30 s cycles on ice). Next, 10 µL of Benzonase and 2 mM MgCl_2_ were added and the sample was incubated for 30 min at 37 °C. The lysate was clarified by centrifugation (18,000 rcf for 30 min at RT). The supernatant was collected and transferred to a protein LoBind tube.

#### 4.5.2. Protein Concentration Estimation and Proteolytic Digestion

The total protein concentration was estimated using the Pierce BCA Protein Assay Kit (Thermo Fisher), using triplicate determination. Next, an aliquot corresponding to 400 µg of total protein was diluted 10-fold with ice-cold ethanol and incubated at −40 °C for 1 h. After centrifugation (12,000 rcf for 30 min at 4 °C), the protein pellet was washed with 1 mL of ice-cold acetone, followed by centrifugation as above. The supernatant was carefully removed and the pellet was dried in a laminar flow hood. Next, the protein pellet was re-solubilized in 50 µL of 6 M GuHCl, followed by dilution to give final concentrations of: 0.2 M GuHCl, 5% acetonitrile, 50 mM ammoniumbicarbonate, 2 mM CaCl_2_, pH 7.8. Trypsin was added in a ratio of 1:20 (*w:w*). The sample was incubated for 14 h at 37 °C under gentle shaking. Digestion was quenched by bringing the solution to a final concentration of 0.5% trifluoroacetic acid (TFA), and the sample was spun down at 18,000 rcf for 5 min at room temperature. A 5-µL aliquot of the supernatant (corresponding to 1 µg of protein) was used for digest control using a monolithic high performance liquid chromatography (HPLC) system, as described previously [64]. The remainder of the sample (399 µg) was desalted using C18 SPEC tips (4 mg, Varian) and a vacuum manifold. The tips were activated with 100 µL 100% acetonitrile (ACN) and equilibrated with 3 × 100 µL 0.1 TFA. The sample was loaded, pulled through the SPEC tip, reloaded, washed with 3 × 100 µL 0.1% TFA, and eluted with 2 × 50 µL 70% ACN. After elution, an aliquot corresponding to 50 µg was transferred to a new tube for the global proteome analysis. Both samples (global proteome and phosphoproteome) were dried under vacuum.

#### 4.5.3. Peptide Fractionation for Global Proteome Analysis

The dried 50-µg digest aliquot was resolubilized in 10 mM ammonium formate, pH 8.0. The sample was fractionated by reversed-phase chromatography at pH 8.0 on a Biobasic C18, 0.5 × 150 mm, 5 µm particle size column, using an UltiMate 3000 LC system (both Thermo Scientific, Germany), with buffer A consisting of 10 mM ammonium formate, pH 8.0, and B consisting of 84% ACN in 10 mM ammonium formate, pH 8.0. Peptides were loaded onto the column in buffer A at a flow rate of 12.5 µL/min, and separated using the following gradient: 3% B for 10 min, 3–45% B in 40 min, 45–60% B in 5 min, 60–95% B in 5 min, 95% B hold for 5 min, 95–3% B in 5 min, re-equilibration of the column with 3% B for 20 min. A total of 20 fractions were collected at 1 min intervals from 5 to 70 min in a concatenation mode and all fractions were completely dried and stored at −80 °C until further use.

#### 4.5.4. Phosphopeptide Enrichment for Phosphoproteome Analysis

The dried 349-µg aliquot was re-solubilized in 1 mL of 80% ACN, 5% TFA, and 1 M glycolic acid (buffer 1) for phosphopeptide enrichment using titanium dioxide (TiO_2_; Titansphere TiO, 5 µm particle size, GL Sciences Inc, Japan), as described previously [65,66]. (1) TiO_2_ beads (6:1 TiO_2_:peptide, *w:w*) were added to the sample, followed by incubation for 10 min on a shaker at room temperature. The sample was centrifuged at 3,000× *g* for 30 s. (2) The supernatant was carefully transferred to a second tube containing TiO_2_ beads (3:1, *w:w*), incubated and centrifuged as before. (3) The supernatant was then transferred to a third tube containing TiO2 beads (1.5:1, *w:w*) incubated and centrifuged as before. The supernatant was carefully removed and discarded, 100 µL buffer 1 was added to the third tube, and the TiO_2_-slurry was transferred to the second tube, mixed and transferred to the first tube, mixed and transferred to a new tube. The sample was centrifuged as before and the supernatant was discarded. Next, the beads were washed with 100 µL buffer 2 (80% ACN, 1% TFA), mixed for 15 s, and centrifuged as before. The solution was discarded, followed by another wash step with buffer 3 (10% ACN, 0.1% TFA). The TiO_2_ beads were dried for 10 min under vacuum, followed by incubation with 100 µL of 1% (*v/v*) ammonium hydroxide (elution buffer), pH 11.3 for 10 min on a shaker at room temperature. The beads were pelleted as before, and the eluate was removed and acidified with 8 µL of 100% formic acid (FA) and 2 µL of 10% TFA. Another 30 µL of elution buffer was added to the beads, mixed for 15 s, centrifuged as before, and the eluates were pooled. Samples were desalted with self-made stage tips, prepared according to Rappsilber et al [67] using C18 3M Empore™ SPE Extraction Disks (Sigma-Aldrich) and 25 µg of OLIGO™ R3 Reversed-Phase resin (Applied Biosystems). Samples were desalted as described above.

#### 4.5.5. Nano-LC-MS/MS in Data Dependent Acquisition Mode

Online LC separations were performed using Ultimate 3000 RSLCnano systems, equipped with a trap column (100 μm × 2 cm, C18 Acclaim Pepmap viper), and a nanoscale analytical column (75 μm × 50 cm, C18 Acclaim Pepmap Viper; all from Thermo Fisher Scientific). All samples were pre-concentrated in 0.1% TFA for 5 min at a flow rate of 20 µL/min, followed by separation on the analytical column at 250 nL/min using a binary buffer (A: 0.1% formic acid; B: 84% acetonitrile, 0.1% formic acid) ranging from 3–35% B in 120 min, at 60 °C. MS survey scans were acquired in the Q Exactive HF (Thermo Scientific) from m/z 300 to 1500 at a resolution of 60,000, using an automatic gain control (AGC) target value of 3 × 10^6^, and a maximum injection time of 120 ms. The 15 most abundant precursors with charge states 2–4 were selected with an isolation width of m/z 1.2, an AGC target value of 5 × 10^4^, and a maximum injection time of 200 ms, fragmented using higher energy collisional dissociation (HCD) with a normalized collision energy of 27, and analyzed at a resolution of 15,000. The dynamic exclusion was set to 20 s.

### 4.6. Discovery Data Search and Analysis

#### 4.6.1. Database generation

Custom sample-specific databases were generated for the discovery-based deep proteomic profiling of the two initial tumor samples based on the obtained WES and RNASeq data. Base protein sequence database was obtained from UniProt (September 2017; 20,218 protein sequences). In the case of identified point mutations leading to nonsynonymous SNVs, all of the potentially affected protein sequence isoforms were obtained from RefSeq (accession date: 15 September 2017) and the respective SNVs were incorporated, using an R script developed in-house. For stop-gain mutations, the respective truncated protein sequence was added, while for stop-loss and frameshift mutations, particular CDS were obtained and (after incorporation of the mutation) the translated variant protein product was added to the custom database. In case multiple protein isoforms were affected, all of the mutated and reference sequences were added to the database.

#### 4.6.2. Database Search Strategies

For global proteome profiling and phosphoproteome analysis, the database searches were performed using Proteome Discoverer 2.2 (Thermo Scientific). The searches were conducted in a target/decoy manner against the sample-specific databases (20,779 and 20,794 target sequences, respectively) plus a common contaminant database (257 target sequences), using the Mascot search algorithms (Version 2.6.1, Matrix Science) and Sequest HT. Precursor mass tolerance was limited to 10 ppm and fragment mass tolerance to 0.02 Da. Cleavage specificity was set to fully tryptic, allowing for a maximum of two missed cleavages. Carbamidomethylation of cysteines was defined as a fixed modification and oxidation of methionine as a variable modification for all searches. Additionally, N-terminal protein acetylation and formation of pyro-glutamic acid from N-terminal glutamine residues were set as variable modifications for global proteome profiling, while phosphorylation of serine, threonine, or tyrosine residues was allowed as variable modification for phosphoproteome analysis. The results were evaluated with Percolator [68] and filtered for 1% false discovery rate (FDR; PSM, Peptide, and Protein level), additionally excluding proteins without uniquely assigned peptides, as well as ones not marked as ‘master’ proteins. The ptmRS node [69] was used in the phosphoRS-mode to evaluate the phosphorylation site localization, setting a minimum confidence level of 90%.

For variant peptide identification, searches were conducted with MSGF+ (v10282), using the SearchGUI interface (Version 3.2.20) [70]. The databases and search settings were the same as described above, but group-specific FDR calculation was utilized. Validation was performed with an R script: First, the search results were parsed using the mzID package, and only variant-specific PSMs were used for FDR calculation, i.e., only PSMs where all protein candidates originated from the variant database where allowed. Target-decoy based FDR calculation was then performed using the open-source MSnID software package [71], utilizing a multi-variant filter optimization (‘Nelder Mead’ method) based on MSGF+ E-score and precursor mass deviation to achieve 1% group-specific FDR on the peptide level.

#### 4.6.3. Quantitative Comparison of the Deep Proteome Data

NSAF values were calculated for all proteins identified at 1% protein FDR with at least one unique peptide. A G12V/WT ratio was calculated for each protein using two NSAF values (NSAF^G12V^/NSAF^WT^). Notably, some proteins were only identified in one of the two tumors. Although these proteins may represent important candidates, ratio calculation is not possible. Therefore, in these cases, missing peptide and PSM values for the protein were set to 1 and 3, respectively, to allow the calculation of a virtual ratio (these values are labeled in Appendix A). To compensate for the potential impact of (i) these imputed NSAF values and (ii) the poor resolution and precision of spectral counting for proteins of low abundance in both samples, only proteins showing at least a 4-fold change between the tumors were considered as differentially regulated.

### 4.7. Development of Parallel Reaction Monitoring Assays for Selected Mutations

#### 4.7.1. Mutation Selection

Peptides were selected to represent all mutations detected with high confidence using group-specific FDR calculations in the initial *KRAS^WT^* and *KRAS^G12V^* proteogenomics analysis, as well as their non-mutated counterparts. Peptides corresponding to the LAMA5 SNV H2036R were a priori excluded from analysis due to unfavorable amino acids composition. The CSPG4 R1842Q SNV-specific peptide DVNERPPQPQASVPLQLTR showed an inconsistent LC-MS/MS response and therefore was deemed unsuitable for quantification. Since the KRAS G12V mutation is located in a conserved sequence region that is shared between wildtype KRAS, NRAS, and HRAS, alternative proteotypic peptides were included in order to enable adequate quantification of the KRAS^G12V^ mutation rate on the protein level.

#### 4.7.2. Synthesis, Purification, and Quantification of Stable Isotope Labeled Standard (SIS) Peptides

The SIS peptides were synthesized in-house using a Syro I synthesis unit (MultiSynTech, Witten, Germany) and Fmoc chemistry and purified as described previously [72]. Heavy-labeled arginine (^13^C_6_
^15^N_4_) and lysine (^13^C_6_
^15^N_2_) were incorporated at the C-terminus. Peptide concentrations were determined using amino acid analysis according to Cohen et al. [73]. Quantification was conducted against a five-point calibration curve of derivatized amino acids, with a concentration range from 5–25 pmol/µL.

#### 4.7.3. Nano-LC-MS/MS in PRM Mode

LC separation of all PRM runs was performed with the same equipment and buffers as described above for discovery proteomics. Samples were preconcentrated for 5 min at a 20 µL/min flow rate, and separated on the nanoscale analytical column at 250 nL/min with a gradient ranging from 5–26% B in 60 min. The Q Exactive HF (Thermo Scientific) was operated in PRM mode at a resolution of 240,000 with a fixed first mass of m/z 150. Target precursor ions were isolated with the quadrupole isolation window set to m/z 0.4 and fragmented with a normalized HCD collision energy of 27. An AGC target of 3 × 10^6^ was used, allowing for a maximum injection time of 250 ms. Data was acquired in time-scheduled mode, allowing a 2-min retention-time window for each target.

#### 4.7.4. PRM Development and Analysis

In order to verify the linearity of response and to determine the limit of blank (LOB), limit of detection (LOD), and lower limit of quantification (LLOQ), response curves of SIS peptides were acquired. First, a background matrix was generated by pooling aliquots of all individual samples, and SIS peptides were spiked in at eight different concentrations, covering a range of three orders of magnitude. Based on prior determination of individual SIS response factors, the highest concentrations varied between 10 fmol and 100 fmol, while the lowest concentrations were between 5 amol and 50 amol on-column. For the analysis of patient samples, a total amount of SIS peptide ranging from 0.1 to 5.0 fmol was spiked into 3 µg of total tissue protein digest, depending on the expected endogenous concentration. Skyline software [74] was used to analyze all of the PRM data. The four to six most suitable transitions of every light and SIS peptide pair were chosen, including at least one diagnostic transition specific for the respective single amino acid variant. All data was manually inspected for correct peak detection, retention time and integration. The peak areas of the light and SIS peptides were exported, and the L/H peak area ratios were used for quantification.

## 5. Conclusions

Although mass spectrometers and mass spectrometric methods are becoming increasingly sensitive and robust, deep proteome profiling for patient phenotyping might be difficult to translate into a clinical setting. The gap between research and the clinic can be partially attributed to: (i) non-automated and relatively long workflows and analysis times; (ii) complicated data analysis; (iii) the high cost of the rather delicate instrumentation; and (iv) the associated need for well-trained and experienced, dedicated personnel.

We therefore sought to evaluate whether targeted MS assays, which can be optimized for throughput and robustness while still providing precise and sensitive quantification of selected candidates, can be used to quantify the selected mutations identified and confirmed in our in-depth proteogenomic profiling experiments. While the setup used in our proof-of-principle experiment is rather expensive and sensitive (nano-LC and a Q-Exactive HF mass spectrometer), these assays can be transferred to less costly triple-quadrupole-based multiple reaction monitoring (MRM) methods using higher flow rates-and therefore more robust chromatography-in a clinical setting. Our data confirms that targeted MS can be used to quantify the actual mutation rates on the protein level, and that this approach is applicable to real-life samples that have not been collected under well-controlled studies, but are indeed used for hotspot sequencing and treatment decisions. This capability might represent a valuable tool for better phenotyping individual tumors and perhaps for explaining the discordance between genomic testing in precision oncology and unexpectedly low response rates in certain cancer types. Our data not only shows that, for *KRAS^G12V^*, there can be a strong disagreement between WES-based variant frequencies (85% vs. 32% for T1) and actual mutation rates on the protein level (between 100% in T2 and 10% in T5), but also shows that protein expression can vary substantially, both between different tumors and between matching tumor and healthy tissue from the same patient.

The T5 patient whose tumor had 90% wildtype KRAS despite the presence of the *KRAS^G12V^* mutation was treated with Bevacizumab to target VEGFR, a common target in mCRC, but deceased 10 months after diagnosis. Knowledge of the rather low expression of mutated KRAS in the patient’s tumor might have allowed a combinatorial anti-EGFR treatment to potentially improve the outcome. While this is rather speculative and impossible to verify, our data presents a potential opportunity how fairly simple targeted-MS assays might help with treatment decisions at comparatively low effort and cost and requiring only minute amounts of sample.

## Figures and Tables

**Figure 1 cancers-11-01907-f001:**
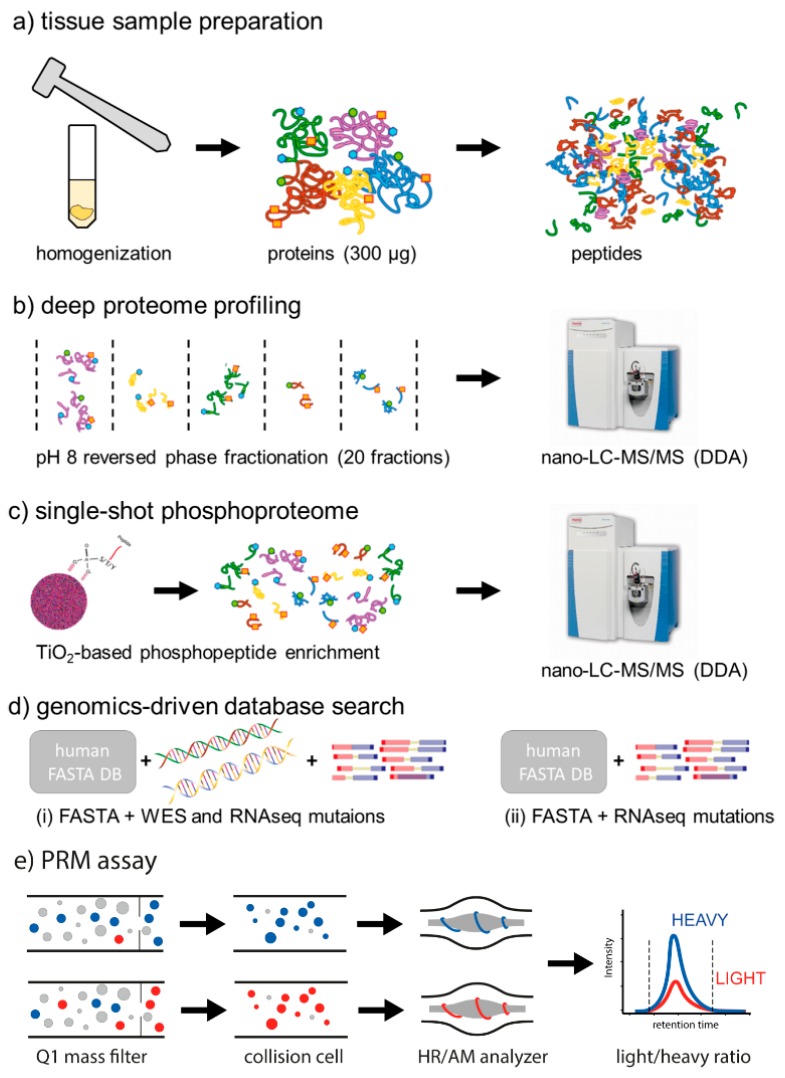
Proteogenomics analysis of human colorectal cancer (CRC) liver metastases. (**a**) Fresh-frozen tissue samples were homogenized using mortar and pestle. An aliquot corresponding to 400 µg of total protein was subjected to protein precipitation using ethanol. After resolubilization in GuHCl buffer, proteins were digested overnight using trypsin. (**b**) For deep proteome profiling, 50 µg of tryptic digest were fractionated by high pH reversed-phase chromatography. A total of 20 concatenated fractions was analyzed by nano-LC-MS/MS on a Q-Exactive HF in data dependant acquisition (DDA) mode. (**c**) For single-shot phosphoproteome analysis, 350 µg of tryptic digest were subjected to TiO_2_-based phosphopeptide enrichment. The eluate was analyzed by nano-LC-MS/MS on a Q-Excactive HF in DDA mode. (**d**) For genomics-driven database search, the Swissprot human FASTA database was complemented with all mutations detected in both the corresponding whole exosome sequencing (WES) and RNAseq data of the same tumor, as well as the mutations detected only by RNAseq. (**e**) A parallel reaction monitoring (PRM) assay was developed for the parallel quantification of eight different mutated peptides and their canonical variants using stable isotope labeled standard (SIS) peptides in a single 60 min nano-LC-PRM run, in order to determine actual mutation rates on the protein level.

**Figure 2 cancers-11-01907-f002:**
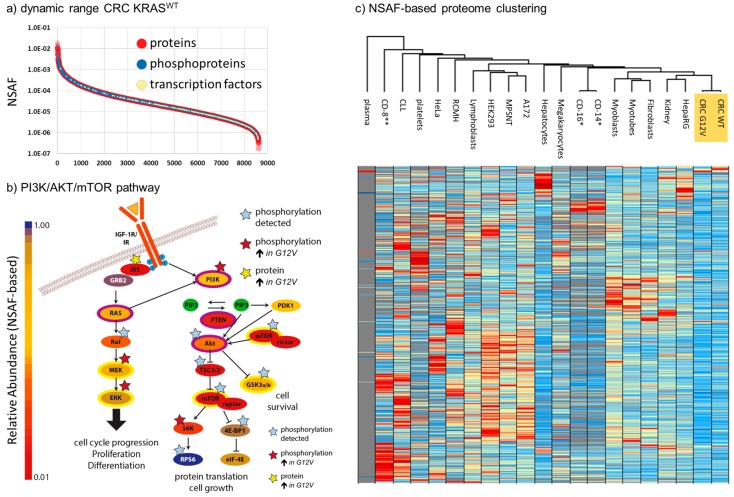
Deep-proteome profile of CRC liver metastases. (**a**) Dynamic range of 8603 unique proteins detected in the *KRAS^WT^* sample. Proteins are ranked by normalized spectral abundance factors (NSAF) values, with high values representing high levels of expression. (**b**) Representation of the PI3K/AKT/mTOR pathway. Relative protein abundance of the represented proteins was determined based on NSAF and is reflected in the depicted color code. (**c**) Proteome-wide comparison of 19 primary tissues/cells and cell models with the CRC tissues (highlighted in orange), based on NSAF. All data is taken from in-house analyses [26,32]. * CD-14 and CD-16 [33] as well as ** CD-8 [34] were reprocessed from recent proteome studies. Only proteins found in one of the CRC samples, and at least 10 other cell/tissue types, are shown.

**Figure 3 cancers-11-01907-f003:**
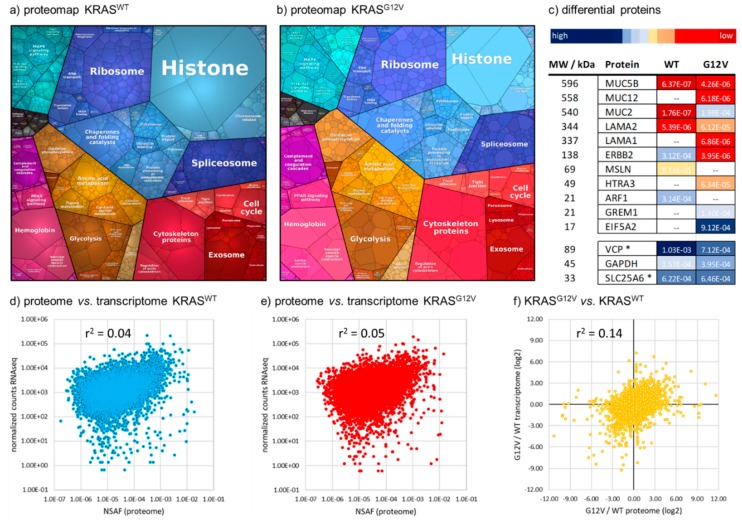
Comparison of *KRAS^WT^* and *KRAS^G12V^* mCRC proteomes. (**a**) Proteomaps [36] representing the quantitative proteome of *KRAS^WT^* and (**b**) *KRAS^G12V^* mCRC tumors based on biological functions. (**c**) Selected proteins having a demonstrated role in CRC and being more than 6-fold differentially regulated between the tumors. Color coding represents relative expression based on NSAF values. GAPDH, VCP, and SLC25A6 were added as ‘loading controls’. – = not detected in the respective tumor; * VCP and SLC26A6 were selected as two proteins with a very stable expression among the reference proteomes, as they showed less than 25% relative standard deviation of their NSAF values among 18 proteomes. (**d**) Poor correlation of proteome and transcriptome data for both the *KRAS^WT^* and (**e**) the *KRAS^G12V^* tumor. (**f**) Poor correlation between G12V/WT ratios calculated based on protein (NSAF, *x*-axis) and mRNA (RNAseq normalized counts, *y*-axis) expression, log2-transformed.

**Figure 4 cancers-11-01907-f004:**
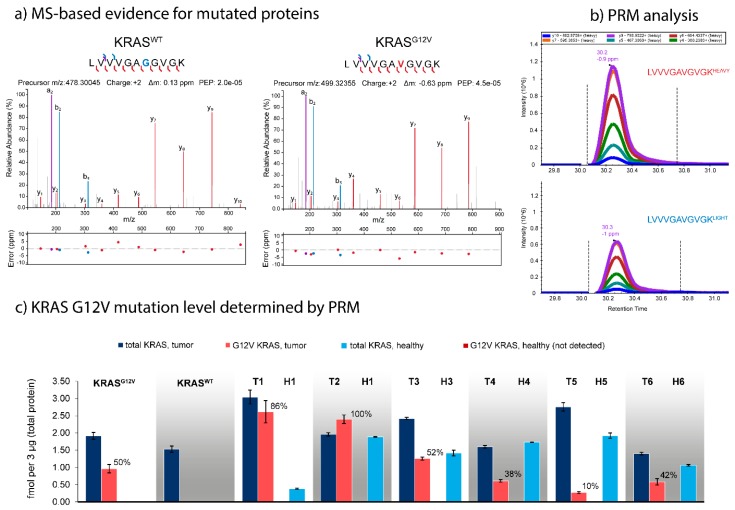
Targeted quantification of protein mutation rates, exemplified for *KRAS^G12V^* and *KRAS**^WT^***. (**a**) High resolution high confidence MS/MS spectra of the peptides LVVVGAGGVGK (KRAS^WT^) and LVVVGAVGVGK (KRAS^G12V^). Spectra were annotated using IPSA [49]. (**b**) Quantification of the LVVVGAVGVGK peptide in the original KRAS^G12V^ tumor sample by nano-LC-PRM with 2.28 fmol of SIS peptide spiked into 3 µg of total tissue protein digest. (**c**) Quantification of total KRAS and KRAS^G12V^ protein expression across multiple mCRC liver as metastases (KRAS^G12V^, KRAS^WT^, well as T1–T6 and their matched healthy tissues H1–H6). KRAS^G12V^ could only be detected in the G12V-positive tumors, whereas the protein mutation rate varied considerably between tumors. T1 shows a massive upregulation of KRAS expression (at 87% mutation rate) compared to H1, whereas T5 shows a very low G12V mutation rate of only 10%.

**Table 1 cancers-11-01907-t001:** Overview of stable- isotope labeled standard (SIS) peptides used for the quantification of mutation rates on the protein level. Protein functions were derived from Uniprot. * The KRAS G12V mutation is located in a region shared between wildtype KRAS, NRAS, and HRAS, thus alternative proteotypic peptides were included.

Gene^mutation^	Protein Name	Function	Variant	SIS Sequence
(Found in)				
*SRPX*2^E234K^	Sushi repeat-containing protein SRPX2	Ligand for the urokinase plasminogen activator surface receptor. Plays a role in angiogenesis. Involved in cellular migration and adhesion.	Canonical	GPEPGSHFPEGEHVIR
(WES + RNAseq)			E234K	GPEPGSHFPK
*KS6A5* ^D554N^	Ribosomal protein S6 kinase alpha-5, S6K-alpha-5, EC 2.7.11.1	Ser/Thr kinase required for the mitogen or stress-induced phosphorylation of transcription factors CREB1, ATF1, and the regulation of the transcription factors RELA, STAT3, ETV1/ER81.	Canonical	DLKPENLLFTDENDNLEIK
(WES + RNAseq)			D554N	DLKPENLLFTNENDNLEIK
			*NRAS **	SYGIPYIETSAK
*KRAS* ^G12V^	GTPase KRas	Ras proteins bind GDP/GTP and possess intrinsic GTPase activity. Plays an important role in the regulation of cell proliferation.	*HRAS **	SFADINLYR
(WES + RNAseq)			*H/N/KRAS **	LVVVGAGGVGK
			Canonical	SFEDIHHYR
			G12V	LVVVGAVGVGK
*PTBP1* ^K508E^	Polypyrimidine tract-binding protein 1	Plays a role in pre-mRNA splicing and in the regulation of alternative splicing events.	Canonical	DYGNSPLHR
(WES + RNAseq)			Canonical	VLFSSNGGVVK
			K508E	VLFSSNGGVVEGFK
*ARL2* ^V141A^	ADP-ribosylation factor-like protein 2	Small GTP-binding protein which cycles between an inactive GDP-bound and an active GTP-bound form	Canonical	EVLELDSIR
(WES)			V141A	EALELDSIR
*PPP1R14C* ^T10A^	Protein phosphatase 1 regulatory subunit 14C	Inhibitor of the PP1 regulatory subunit PPP1CA.	Canonical	SVATGSSEATGGASGGGAR
(WES)			T10A	SVATGSSEAAGGASGGGAR
*HAUS7* ^T244A^	HAUS augmin-like complex subunit 7	Contributes to mitotic spindle assembly, maintenance of centrosome integrity and completion of cytokinesis as part of the HAUS augmin-like complex.	Canonical	TEYFAQHEQGAAAGAADISTLDQK
(WES + RNAseq)			T244A	AEYFAQHEQGAAAGAADISTLDQK
*TBD2B* ^A8G^	TBC1 domain family member 2B	May act as a GTPase-activating protein.	Canonical	AEEGGGGGEGAAQGAAAEPGAGPAR
(WES)			A8G	GEEGGGGGEGAAQGAAAEPGAGPAR

WES: whole exosome sequencing.

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
