# Peer review of "Proteogenomics of Colorectal Cancer Liver Metastases: Complementing Precision Oncology with Phenotypic Data"

_cancers, 2019, doi:10.3390/cancers11121907_

Round 1

Reviewer 1 Report

The manuscript describes a proteogenomic strategy to study somatic mutations in tumors, KRAS wildtype and KRAS G12V mutation. With eight mutation candidates, they developed a PRM assay to quantify the mutated and canonical protein variants. This study may help the improvement of cancer treatment in precision oncology. It may be interesting and informative to other researchers. However, there are some technical issues to be addressed. See below for specific comments.

The study used label free quantification strategy to compare KRAS wildtype and KRAS G12V tumors. It is not described clearly that it has only one time experiment or multiple repeats. How is the data reproducibility? Same issue for the Ten mutation peptides were identified with the database search using WES and RNAseq results. It will be useful if authors present the proteogenomics identified mutation peptides is covered by WES, RNAseq, or both. If the peptide was not covered by RNAseq, it might be not a strong candidate. PAGE 13, line 480, for acetone precipitation, it is not a common protocol that use 10-fold with ice-cold ethanol, and incubated at 40 oC for 1h. PAGE 13, in “protein concentration estimation and proteolytic digestion”, the authors measured protein concentration before acetone precipitation. Therefore, the digested protein amount is not as the same as described in the manuscript, because protein loss during precipitation step. Normally, people estimates protein concentration after protein acetone precipitation and pellet re-solubilized. Figure 2a shows the dynamic range of the proteins. But the dots with different color are not separated from each other. It is better if the author can change another way to show the data or just delete the phosphoproteins and transcription factors information. The phosphoproteomics data is not shown in the supplemental table. The phsophoproteomics data quality seems not very high, and it was not fully analyzed and well presented in the manuscript.

Author Response

Reviewer #1 --- Comments and Suggestions for Authors

The manuscript describes a proteogenomic strategy to study somatic mutations in tumors, KRAS wildtype and KRAS G12V mutation. With eight mutation candidates, they developed a PRM assay to quantify the mutated and canonical protein variants. This study may help the improvement of cancer treatment in precision oncology. It may be interesting and informative to other researchers. However, there are some technical issues to be addressed. See below for specific comments.

>>>> We thank this reviewer for the positive comments.

The study used label free quantification strategy to compare KRAS wildtype and KRAS G12V tumors. It is not described clearly that it has only one time experiment or multiple repeats. How is the data reproducibility?

>>>> The discovery part of the study is based on a single technical replicate per tumor, while the PRM part of the manuscript is based on triplicate analyses. From our own experience, NSAF (normalized spectral abundance factor)-based quantification is a robust strategy -- especially for samples that are more heterogeneous than those from simple cell culture experiments. For this reason, we preferred NSAF to conducting, for example, a MaxQuant-based Label-free Quantitation experiment using precursor areas and multiple fractions. In order to reduce the impact of a potentially low precision derived from spectral counting, especially on the lower abundance proteome, we used rather strict criteria to consider a protein as “regulated”. We therefore think that the overall changes we see are reproducible, as we did not consider small fold-changes to be relevant. In addition, we compared the NSAF-based quantification with the PRM-based quantification using stable isotope labeled standard peptides, which is the most precise MS-based quantification strategy (see section “Robustness of NSAF-based relative quantification” and suppl. figure 1). Both strategies correlate very well – much better than we expected. We therefore think that the reproducibility of the data is much less of a problem than the inherently heterogeneous nature of tumor samples (and their sampling).

Same issue for the Ten mutation peptides were identified with the database search using WES and RNAseq results. It will be useful if authors present the proteogenomics identified mutation peptides is covered by WES, RNAseq, or both. If the peptide was not covered by RNAseq, it might be not a strong candidate.

>>> This information is actually embedded in supplemental table 1, but we understand that this is far from ideal. We, therefore, added this information in the revised version of table 1 (see below).

Interestingly, only ARL2 V141A, PP1R14C T10A and TBD2B A8G were only detected by WES, all other mutations by both WES and RNAseq.  Notably, PP1R14C T10A was detected in 5 of the analyzed tumors --- but not its wildtype form --- despite being only detected by WES, but not RNAseq. Whereas TBC12DB A8G could only be quantified in two tumors, and indeed might not be a good candidate. 

PAGE 13, line 480, for acetone precipitation, it is not a common protocol that use 10-fold with ice-cold ethanol, and incubated at 40 oC for 1h.

>>> This is probably a misunderstanding. Indeed, we conducted Ethanol precipitation, not Acetone precipitation. Our protocol, however, includes an Acetone Wash of the protein pellet in order to remove potentially contaminating lipids, etc. For Ethanol precipitation this is a standard procedure.

PAGE 13, in “protein concentration estimation and proteolytic digestion”, the authors measured protein concentration before acetone precipitation. Therefore, the digested protein amount is not as the same as described in the manuscript, because protein loss during precipitation step.

Normally, people estimates protein concentration after protein acetone precipitation and pellet re-solubilized.

>>> We have optimized our Ethanol precipitation protocol extensively over the past years and used in numerous projects. Even for small sample amounts down 5 µg of total protein, we have always achieved recoveries of 90% and more and the variation in protein amount between pre- and post-precipitation is typically lower than the accuracy of the BCA assay. Here, we conducted the BCA assay prior to protein precipitation in order to guarantee that the samples to be compared are processed using the same amount of starting material. Thus, we ensured to keep sample preparation right from the beginning as comparable as possible. It might be more comprehensive to include a second protein estimation after the precipitation, but given the generally good recoveries in our hands, we did not feel an urgent need to do so. Importantly, we quality controlled all digested samples for their digest efficiency and reproducibility using monolithic HPLC columns with UV detection and the UV traces we obtained for the samples confirmed both, (i) that the sample preparation was reproducible and (ii) that the total amount of digested peptide was as expected. In fact these are essential criteria for us to move to the next level, either discovery or targeted analysis of any sample.

Figure 2a shows the dynamic range of the proteins. But the dots with different color are not separated from each other. It is better if the author can change another way to show the data or just delete the phosphoproteins and transcription factors information.

>>> Our idea was to demonstrate that the two relevant classes, phosphoproteins and transcription factors, were actually covered across the entire dynamic range of the proteome. We tried to present the data in a way that the “whole proteome dots” in blue are larger than those of the 2 other classes, so that all of them should be visible at the same time. We tried to improve this visualization in the revised version (see “new” below), however, we wanted to keep the overall concept to plot all of them in the same graph (see below).

The phosphoproteomics data is not shown in the supplemental table. The phsophoproteomics data quality seems not very high, and it was not fully analyzed and well presented in the manuscript. 

>>> We agree with the reviewer, that the phoshoproteome data were indeed underrepresented in the manuscript. The reason for this is, that apart from the general heterogeneity of tumors and the aforementioned sampling, the phosphoproteome is extremely labile to external stimuli and, as for instance shown by CPTAC, also minimal differences in the way how samples have been treated/stored right after surgery --- considerably more than the more stable proteome. With the experience of numerous quantitative phosphoproteomics (and other PTM-centric) studies we have conducted in recent years (e.g. Beck et al., Blood 2014; Beck et al., Blood 2017; Plenker et al., Sci Transl Med 2017; Solari et al., Mol Cell Proteomics 2016; Di Conza et al., Cell Rep 2017; Gonczarowska-Jorge et al., Anal Chem 2017; Shema et al., Mol Cell Proteomics 2018; Mnatsakanyan et al., Nat Commun 2019), we did not feel that the data is sufficiently robust to allow a thorough comparison as we have done for the proteome and therefore refrained from over-interpreting the delicate phosphoproteome of the tumors. We rather aimed at providing some ideas of what is feasible without conducting large-scale studies based on TMT-multiplexing as has been shown by CPTAC and others. Importantly, the combination of both, single-step enrichment followed by PRM/MRM to “absolutely” quantify not only mutation rates but also phosphorylation states is an important option for the future --- again to phenotype individual tumors in the clinic, not only for conducting large-scale studies.

Reviewer 2 Report

In this manuscript, the authors used targeted mass spectrometry to determine the impact of KRAS mutation on the protein level so that it could provide information on the level of wild type and G12V mutated protein levels and thus if the patient would be a suitable candidate for EGFR targeted therapies in colorectal cancer.  They also compared genomic data with proteomic data showing the divergence that can occur between gene and protein expression.  They developed a parallel reaction monitoring assay to quantify the amount of mutated versus wild type KRAS protein.

Q1

Why only n=1 for the screening samples for proteomics?

Given issues with tumour heterogeneity, it is very hard to understand why one would carry out such comprehensive proteomic studies on any sample number of less than three for each variable.  It appears from the huge amount of additional work done, that the authors are trying to show how these results are not as a result of contamination from other tissues.  If there were only very limited samples, that should be acknowledged.  Tumour heterogeneity should also be acknowledged.

Q2

If only 13% of wild type KRAS respond to anti-EGFR targeted therapies, why only compare KRAS in the PRM assays?

Q3

For the tumour and normal samples that were used in the PRM assays, is there any follow up data as to how the patients responded to treatment?

Q4

From the other targets suggested, do the authors feel that any of those could also be used as a means for screening patients that would be suitable for specific treatments?  Also could any of these targets be used in conjunction with the KRAS status to provide more information on deciding regime therapies?

Author Response

Reviewer #2 --- Comments and Suggestions for Authors

In this manuscript, the authors used targeted mass spectrometry to determine the impact of KRAS mutation on the protein level so that it could provide information on the level of wild type and G12V mutated protein levels and thus if the patient would be a suitable candidate for EGFR targeted therapies in colorectal cancer.  They also compared genomic data with proteomic data showing the divergence that can occur between gene and protein expression.  They developed a parallel reaction monitoring assay to quantify the amount of mutated versus wild type KRAS protein.

Q1

Why only n=1 for the screening samples for proteomics?

Given issues with tumour heterogeneity, it is very hard to understand why one would carry out such comprehensive proteomic studies on any sample number of less than three for each variable.  It appears from the huge amount of additional work done, that the authors are trying to show how these results are not as a result of contamination from other tissues.  If there were only very limited samples, that should be acknowledged.  Tumour heterogeneity should also be acknowledged.

>>> This is an important and just comment. Indeed, the enormous heterogeneity of tumors is even multiplied by the way how samples are collected in the clinic and how they are stored. As this reviewer pointed out -- and in contrast to other studies that focus on seeing common patterns across large cohorts -- we aimed at analyzing and phenotyping individual tumors rather than cohorts and ensuring that we can distinguish the tumor proteome from artifacts (which in larger studies is done based on pattern recognition approaches). A direct consequence of this strategy is the development of targeted assays to phenotype individual tumors, a strategy that we aim to apply on real patient samples that are not derived from studies under well-controlled conditions and SOPs, as demonstrated here.

We added this statement (underlined) to the “Conclusions” section in order to better emphasize this rationale:

“Our data confirms that targeted MS can be used to quantify the actual mutation rates on the protein level, and that this approach is applicable to real-life samples that have not been collected under well-controlled studies, but are indeed used for hotspot sequencing and treatment decisions”. 

Q2

If only 13% of wild type KRAS respond to anti-EGFR targeted therapies, why only compare KRAS in the PRM assays?

>>> This reviewer raises a clinically important point. However, in this study we wanted to demonstrate a potential benefit in quantifying mutation levels on the protein level, rather than looking only at the genomics data. We focused on the KRAS G12V mutation as this is the most prevalent mutation in CRC tumors (>10% of tumors) and is supposed to have a direct implication on treatment. Currently, the mere presence of mutated KRAS detected using genomics assay is used clinically to exclude patients from anti-EGFR therapy. We therefore did not aim at quantifying KRAS in wildtype tumors. Though the general KRAS expression level may have an effect on the treatment even for KRAS wildtype tumors, the low response rate to anti-EGFR treatment may rather be a consequence of other mutations within the RAS/MEK/ERK pathway, or as stated in our manuscript:

“Modest response rates in precision oncology can for instance arise from therapeutic resistance due to the activation of alternative signalling pathways.  This has been demonstrated for Bevacizumab, where Vascular endothelial growth factor (VEGF) inhibition can trigger signalling through IFG1R, PDGFR, FGFR, or MET [16].“

We also corrected the share of non-responders from 13% (taken from an older study that focused only one specific anti-EGFR drug) to <30%, which is more in accordance with current knowledge. We apologies for this and corrected this is the revised version of the manuscript.

Q3

For the tumour and normal samples that were used in the PRM assays, is there any follow up data as to how the patients responded to treatment?

>>> There is no clinical follow up related to anti-EGFR therapy since these patients are excluded from this treatment, based on the genomic assay. Also, the patients that were solely analyzed by PRM were not part of a study but part of our internal biobank. As a consequence, the available information is not-standardized and incomplete, which unfortunately is still not unusual in daily routine in the clinic. From what we know, 4 out of the 6 patients whose samples were analyzed from the biobank have passed away after receiving different types of treatment including chemotherapy and/or Bevacizumab (targeting VEGF-A) for several cycles. These results, unfortunately, are in line with poor prognosis of stage III and IV CRC patients with liver metastases. We added a short statement (underlined) about this to the results section of the revised manuscript:

“To further evaluate the utility of our assays, we obtained additional mCRC liver-metastasis samples (T1-T6) and matched healthy liver tissue (H1-H6) from another 6 patients with KRASG12V-positive tumors, as defined by hotspot mutation testing for the presence of selected mutations. Notably, these samples were not part of a well-designed study, but where ‘real-life’ biopsies from our internal biobank and therefore represent the ideal setting to evaluate whether targeted MS can be used to improve the phenotyping of samples that were not collected using specific SOPs for proteome analysis. All these patients were ineligible for targeted anti-EGFR treatment and received different (combinations) of treatment, including anti-VEGF-A treatment with Bevacizumab as well as chemotherapy for several cycles. In line with the poor prognosis of patients with stage III and stage IV CRC, 4 out of the 6 patients from the biobank have passed away.”

Q4

From the other targets suggested, do the authors feel that any of those could also be used as a means for screening patients that would be suitable for specific treatments?  Also could any of these targets be used in conjunction with the KRAS status to provide more information on deciding regime therapies?

>>> We indeed believe that there are other targets to screen patients for specific treatments. This is why we conducted this study as a demonstration project for one genomic test that is used in the clinic to make treatment decisions. Even mutated KRAS itself may be a useful target, besides its implications for anti-EGFR treatment. For example, recent data show that some clinical activity of antibodies designed to target specific KRAS mutants may ultimately depend on the level of that mKRAS in the tumor (Ostrem an Shokat, Direct small-molecule inhibitors of KRAS: from structural insights to mechanism-based design Nature Reviews Drug Discovery volume15, pages771–785 (2016)).
We, however, want to point out that in this proof-of-concept study we focused on mutations we could detect on the proteome level and that were “proteotypic”, which means they were represented by unique peptides with specific characteristics that are ideal for MS-based quantification (e.g. no Met in the sequence, not too short/long, good compatibility with peptide synthesis and purification, etc.).

Round 2

Reviewer 1 Report

The authors have made a genuine and thorough effort to respond to the points I raised on the first version of the manuscript.